Odor descriptive ratings can predict some odor-color associations in different color features of hue or lightness

Tamura Kaori 1 k-tamura@fit.ac.jp
Okamoto Tsuyoshi 2
1 Department of Information and Systems Engineering, Fukuoka Institute of Technology , Fukuoka , Japan
2 Faculty of Arts and Science, Kyushu University , Fukuoka , Japan
Fogt Nick
Electronic publication date: 2023 Apr 20
Publication date: 2023
Volume: 11
Electronic Location ID: e15251
Received 2022 Nov 22; Accepted 2023 Mar 28
Copyright: © 2023 Tamura and Okamoto
Copyright year: 2023
Copyright holder: Tamura and Okamoto
License: This is an open access article distributed under the terms of the Creative Commons Attribution License, which permits unrestricted use, distribution, reproduction and adaptation in any medium and for any purpose provided that it is properly attributed. For attribution, the original author(s), title, publication source (PeerJ) and either DOI or URL of the article must be cited.
License URL: https://creativecommons.org/licenses/by/4.0/

Keywords: Vision-olfactory modulation, Multimodal input, Association, Olfactory impression

Funding: 2019 QR Program (Qdai-jump Research Program) Wakaba 405 Challenge 01296 2022 Research Support Program for Young Scientists in Fukuoka Institute of Technology This study was supported by the 2019 QR Program (Qdai-jump Research Program) and Wakaba 405 Challenge (reference number: 01296), and the 2022 Research Support Program for Young Scientists in Fukuoka Institute of Technology. The funders had no role in study design, data collection and analysis, decision to publish, or preparation of the manuscript.

==============================
Background

Olfactory information can be associated with color information. Researchers have investigated the role of descriptive ratings of odors on odor-color associations. Research into these associations should also focus on the differences in odor types. We aimed to identify the odor descriptive ratings that can predict odor-color corresponding formation, and predict features of the associated colors from the ratings taking into consideration the differences in the odor types.

Methods

We assessed 13 types of odors and their associated colors in participants with a Japanese cultural background. The associated colors from odors in the CIE L*a*b* space were subjectively evaluated to prevent the priming effect from selecting color patches. We analyzed the data using Bayesian multilevel modeling, which included the random effects of each odor, for investigating the effect of descriptive ratings on associated colors. We investigated the effects of five descriptive ratings, namely Edibility, Arousal, Familiarity, Pleasantness, and Strength on the associated colors.

Results

The Bayesian multilevel model indicated that the odor description of Edibility was related to the reddish hues of associated colors in three odors. Edibility was related to the yellow hues of colors in the remaining five odors. The Arousal description was related to the yellowish hues in two odors. The Strength of the tested odors was generally related to the color lightness. The present analysis could contribute in investigating the influence of the olfactory descriptive rating that anticipates the associated color for each odor.

Introduction

Olfactory information can provide specific visual information. Some odors are associated with specific colors, such as a lemon-like odor for yellow and a cherry odor for red (Zellner, 2013). These odor-color associations have attracted attention and have been extensively investigated (Gilbert, Martin & Kemp, 1996; Guerdoux, Trouillet & Brouillet, 2014; Kemp & Gilbert, 1997). Dematte, Sanabria & Spence (2006), using an implicit association test, found that some odor-color pairs demonstrated both association (e.g., strawberry smell and pink) and disassociation (e.g., strawberry smell and turquoise). This report raised the possibility of an implicit association between some odors and colors. Another report indicated that color information can facilitate speeded discrimination of related odors, even when the participants were instructed to ignore the visual information (Dematte, Sanabria & Spence, 2009). In some cases of odor-color pair, an odor presentation can inhibit visual attention of the associated color. We reported that working memory on orange colors were specifically inhibited under the presentation of decanal, which was a citrus odor included in citrus peels (Tamura, Hamakawa & Okamoto, 2018). The orange colors represent the olfactory source, citrus peels, and this representation could suggest that the specific decline of orange-color working memory was probably interpreted by odor-color associations. If the mechanism on the association of odor-color pair is revealed, visual attention or olfactory detection would be modulated based on the knowledge of the associated mechanism.

Many studies have investigated how some odors were associated with specific colors, but unified interpretation has not been established yet. In the review of crossmodal correspondences (but for visual and auditory correspondences), Spence (2011) classed three types of correspondences: structural correspondences resulting from neuronal peculiarities, statistical correspondences based on learning from the environment, and semantically mediated correspondences owing to linguistic terms. Semantic attributes and learning from experiences have been the main focus in crossmodal studies of vision and olfactory. For the semantic attributes, the influence on odor-color association of nameability, familiarity (Nehmé et al., 2016; Stevenson, Rich & Russell, 2012) and the edibility (Maric & Jacquot, 2013) have been investigated. Some other descriptive ratings, such as food-related factors, intensity ratings, or masculinity/femininity of odors, are associated with corresponding colors (Kemp & Gilbert, 1997; Maric & Jacquot, 2013; Zellner et al., 2008). These studies support the hypothesis that odor descriptions, such as edibility, intensity, and food-related factors, can mediate odor-color associations. The odor-color associations could be influenced by learning and experiences, such as learning to associate color and taste (Stevenson, Boakes & Wilson, 2000). Moreover, the cultural backgrounds (Jacquot et al., 2016; Levitan et al., 2014; Nehmé et al., 2016) influence the odor-color association. Considered together, the odor-color association could be affected by semantic consistency, learning experience, and cultural contexts.

Other reports suggested the effect of linguistic system. Western language speakers tend to represent smells as labeling of the olfactory source; however, individuals have abstract terminology for odors in Southeast Asia (Majid & Burenhult, 2014). de Valk et al. (2017) compared Maniq (Southeast Asian minorities) individuals with abstract smell terms and Western language speakers (Thai and Dutch participants). They suggested that when individuals have different language systems for olfaction, their strategy to choose associated colors with smell would be different. Therefore, individuals with similar cultural backgrounds and linguistic systems possibly share several common odor-color associations.

Although we can assume that some olfactory descriptive evaluations would relate to associated colors among individuals with common culture, predicting the association and color characteristics based on the descriptive ratings is challenging. If we are able to determine the influence of olfactory description on the features of associated colors, it would contribute in revealing the mechanism of the odor-color association formation. Our main questions are: (1) which olfactory description can be useful to expect associated colors, and (2) how descriptive ratings of odors will influence the hue and lightness of the associated colors. To solve these problems, a new prediction method should be developed to predict the contribution of odor descriptive ratings for the association of specific colors.

This study aimed to investigate the olfactory descriptive ratings that would be useful in expecting an association color for each odor, and predicting the influence of the ratings on hue and lightness of the associated colors. Even when the individuals shared common background culture, the relationship between the olfactory descriptive ratings and color association could vary across the odor types. Herein, our model aimed to determine the influence of olfactory description accounting for the differences in the odor types, and not to determine the general feature among all the odors to predict the associated colors.

We assessed 13 types of odors, their associated colors, and their descriptive evaluations using the aforementioned methods. The tested odors were rated using five descriptive ratings as follows: Edibility, Arousal, Familiarity, Pleasantness, and Strength. To predict the effect of the descriptive attributes, we introduced a Bayesian multilevel regression model with the data of associated colors for the assessed odors to avoid type-1 error due to unnecessary multiple comparisons (Gelman, Hill & Yajima, 2012).

Materials and Methods

The local ethics committee of the Faculty of Arts and Science, Kyushu University approved the current experiment. All procedures were performed in accordance with the approved guidelines of the local ethics committee of the Faculty of Arts and Science, Kyushu University (#201907). All participants provided written informed consent in accordance with the tenets of the Declaration of Helsinki before participation.

Participants

We recruited 23 university students for this study (women, n = 11; men, n = 12; and mean age ± SD = 20 ± 1.6 years). None of the participants displayed color vision deficiencies following the Ishihara’s test. Based on self-reports, none of the participants had olfactory or neurological deficits. As the odor-color association depends on cultural and language backgrounds (Levitan et al., 2014; Majid & Burenhult, 2014; Majid et al., 2018), we ensured that all participants were native Japanese speakers and from Japanese cultural backgrounds. The participants were instructed to stop eating and drinking 1 h before the experiment; however, they were permitted to drink water.

Environment and display settings

All visual stimuli were presented on a laptop computer screen (MacBook Pro 13-inch, Retina display, resolution 2,560 × 1,600) using Psychtoolbox-3 (Brainard, 1997; Kleiner, Brainard & Pelli, 2007) programmed in MATLAB (MathWorks, Inc., Natick, MA, USA). The monitor was calibrated following standard methods (Brainard, Pelli & Robson, 2002) using a Chroma Meter (CS150; Konica Minolta, Inc., Tokyo, Japan). We placed the laptop computer display on a portable photography box measuring 60 × 60 cm to control the lightness of the environment (Fig. 1A). The backdrop was black and there was no light-emitting diode inside the box during the experiments.

Figure 1 Experiment design and environment.

(A) Experiment procedure; (B) experiment environment and a 20-mL vial containing 15 µl of the odorant solution; and (C) procedure to respond to the associated colors and descriptive ratings after sniffing an odorant; (D) the circle in the CIE L*a*b* color space to determine the color hue from cursor rotation. The labels around the circle’s perimeter indicate color names generally called. First, the center rectangle’s color hue was changed by rotating the cursor. The color hue was determined by the cursor angle θ and the circle in the color space shown in (D). Second, the lightness was selected based on three steps. Third, after defining the associated color, the participants provided descriptive ratings using the visual analog scale.

Odor stimuli

We assessed 13 odors in this study (Table 1). According to a previous report, they were diluted to control the odor intensity, which was almost equivalent (Bushdid et al., 2014).

Table 1 The odorants and concentrations with solvents.

Odorants	Concentration (%)	Solvents	
Trimethyl amine (30%)	0.025	Water	
Butanoic acid	1.000	Water	
4-ethyl-2-methoxyphenol	0.100	1,2-propanediol	
2-ethyl pyrazine	0.400	Mineral oil	
2,3-dimethylpyrazine	0.200	1,2-propanediol	
Limonene	5.000	Mineral oil	
Strawberry aldehyde (ethyl 3-methyl-3-phenylglycidate)	1.000	1,2-propanediol	
4-methyl-3-penten-2-one	1.000	1,2-propanediol	
Butyl acetate	1.000	1,2-propanediol	
(1S)-(-)-α-pinene	15.000	Mineral oil	
Hydroxy citronellal	50.000	Mineral oil	
Isovaleric acid	0.010	Mineral oil	
Propane-1-thiol	0.001	1,2-propanediol	

Five microliters of the odorant solution prepared according to Table 1 was added dropwise to 0.5 × 0.5 cm of cotton wool. Three cotton wool samples with odorant drops were encapsulated in a 20 mL vial (Fig. 1B), and the solution was volatilized for at least 30 min. Immediately before the participant sniffed the sample, the cotton wool was removed and only the odor gas was presented. We counterbalanced the order of presentation of the odorants among the participants. The same 13 odorants were used in both the experiments; the order of presentation was different within the two experiments. We also changed the order of presenting the odorants among the participants. In addition, the participants were not notified that “the odor substances used in the first and second experiments were identical.” The substance name was not written on the vial, and the odor was presented with care such that the participants could not predict the type of odor presented.

Experiment

The participants were instructed to sniff 13 different odorants and indicate the color and semantics associated with each odor. The odor was enclosed in a bottle; the participants opened the bottle upon instruction and sniffed it. Immediately after completing the associative color/semantics evaluation response, the participants sniffed coffee beans for 10 s. Following a 2-min break, they smelled the subsequent odorant (Fig. 1A). The room was ventilated during the break. All participants completed the task twice on different days.

We requested the participants to respond to an associated color with a presented odor and to rate several descriptions. The procedure and color settings were determined in a previous study (Tamura, Hamakawa & Okamoto, 2018). Each participant was handed a vial containing the gaseous odor and asked to sniff the odor. Consequently, the color associated with the odor was reported. First, the participants were asked concerning the hue angle of the associated color and they rotated a cursor corresponding to the hue angle. The angle between the cursor and the center of the circle determined the hue value of the square. The square was presented at the center of the display, and the color was subsequently changed following the cursor movement (Tamura, Hamakawa & Okamoto, 2018). The colors were determined within a circle in the CIE L*a*b* color space, with a center space at L* = 70, a* = 20, b* = 38, and a radius of 60 (Zhang & Luck, 2008) (Fig. 1D). They clicked upon deciding that the current color hue was most associated with the presented odor. Subsequently, we presented three colors with different lightness values of the selected color (L* = 40, 70, and 100). The participants selected the lightness of the color that was most associated with the odor (Fig. 1C).

After deciding the associated color, they were instructed to complete the responses to five descriptor items as follows: Strength, Pleasantness, Familiarity, Edibility, and Arousal, using the visual analog scale (VAS) ranging from 0 (completely different) to 1 (very applicable). For example in the strength evaluation, 1 indicates very strong, 0 indicates very weak, and 0.5 indicates moderate strength. Strength and Edibility would relate to primitive odor evaluation. Pleasantness and Familiarity would reflect basic subjective evaluation for odors, and Arousal would relate to physiological states. The relationship of color-odor association with descriptor items of Strength and Edibility have been reported previously (Kemp & Gilbert, 1997; Maric & Jacquot, 2013). To investigate the influence by different descriptor items, Pleasantness, Familiarity, and Arousal were added. Pleasantness and Familiarity have been used in related olfactory studies with subjective ratings (e.g., Delplanque et al., 2015; Ferdenzi et al., 2017, 2013; Huisman & Majid, 2018). These terms were also used for color-odor association studies (Nehmé et al., 2016). Arousal ratings can be informative to show the participants’ affective states, which has been reported previously Arousal ratings to odors reportedly correlated with the autonomic system (Bensafi et al., 2002).

Our method attempted to measure the colors associated with odors to prevent visual priming effect while choosing associate colors. Conventional studies assessed the selection based on several color patches after sniffing the odors (de Valk et al., 2017; Gilbert, Martin & Kemp, 1996; Levitan et al., 2014; Majid & Burenhult, 2014; Majid et al., 2018). We were concerned that the color patch presentations may could result in the priming of responses since visual priming can support olfactory identification and odor-naming (Gottfried & Dolan, 2003; Guerdoux, Trouillet & Brouillet, 2014; Morrot, Brochet & Dubourdieu, 2001; Olofsson & Gottfried, 2015; Robinson, Reinhard & Mattingley, 2015; van Beilen et al., 2011).

Bayesian estimation

For the statistical analysis, we performed Bayesian estimation with several models. The models were fitted using the R environment (ver.3.4.0) and RStan (ver.2.2.1) with the Markov chain Monte Carlo (MCMC) method. All estimates were made with 3,000 samplings, running four chains to generate random numbers, and a burn-in period of 1,000. We used the Gelman-Rubin statistics R^ to determine if the MCMC estimation converged for all estimation parameters. R^ is generally considered to converge as it approaches 1.10, and each model fit produces R^ < 1.10.

Bayesian multilevel regression analysis

We used Bayesian multilevel regression models to estimate the effects of the following five descriptor items on color responses: Strength, Pleasantness, Familiarity, Edibility, and Arousal. The multilevel models estimated the L*-, a*-, and b*-axis values using odor-level effects. The values of the L*-, a*-, and b*-axes were divided by 100 for normalization. The Bayesian multilevel regression did not require concern about type-1 error from multiple comparisons because the model did not assume a null hypothesis (Gelman, Hill & Yajima, 2012).

We modeled the response value of the a*-axis for the associated colors by each odor as a normal distribution as follows:

a[j]∗∼Normal(α0[i]+αS[i]Strength[i]+αP[i]Pleasantness[i]+αF[i]Familiarity[i]+αE[i]Edibility[i]+αA[i]Arousal[i],σa[i]),

σa[i]>0

where i indicates the odor ID and j denotes the data index. The intercept ( α0[i]) and each coefficient, αX[i], followed a normal distribution with mean coefficients as follows:

α0[i]∼Normal(α0_all,σα0),

αS[i]∼Normal(αS_all,σαS),

αP[i]∼Normal(αP_all,σαP),

αF[i]∼Normal(αF_all,σαF),

αE[i]∼Normal(αE_all,σαE),

αA[i]∼Normal(αA_all,σαA),

where αX_allindicates the average coefficient across all odorants. The prior distributions for the parameters of the standard deviations were as follows:

σα0∼Normal(0,5)σα0>0,

σαS∼Normal(0,5)σαS>0,

σαP∼Normal(0,5)σαP>0,

σαF∼Normal(0,5)σαF>0,

σαE∼Normal(0,5)σαE>0,

σαA∼Normal(0,5)σαA>0.

Regarding the model for a* value estimation, we modeled the b*-axis values as follows:

b[j]∗∼Normal(β0[i]+βS[i]Strength[i]+βP[i]Pleasantness[i]+βF[i]Familiarity[i]+βE[i]Edibility[i]+βA[i]Arousal[i],σb[i]),

where each coefficient followed normal distribution with mean coefficients as follows:

β0[i]∼Normal(β0_all,σβ0),

βS[i]∼Normal(βS_all,σβS),

βP[i]∼Normal(βP_all,σβP),

βF[i]∼Normal(βF_all,σβF),

βE[i]∼Normal(βE_all,σβE),

βA[i]∼Normal(βA_all,σβA),

where βX_all, indicates the average coefficient across all odorants. The prior distributions for the parameters of standard deviations in our model were as follows:

σβbn∼Normal(0,5),

where σβX, > 0.

Similar to the steps with a* and b* values, we model the L-axis values as follows:

L[j]∗∼Normal(λ0[i]+λS[i]Strength[i]+λP[i]Pleasantness[i]+λF[i]Familiarity[i]+λE[i]Edibility[i]+λA[i]Arousal[i],σL[i]),

and the prior distributions for parameters were as follows:

λ0[i]∼Normal(λ0_all,σλ0),

λS[i]∼Normal(λS_all,σλS),

λP[i]∼Normal(λP_all,σλP),

λF[i]∼Normal(βF_all,σλF),

λE[i]∼Normal(βE_all,σλE),

λA[i]∼Normal(αA_all,σλA),

σλn∼Normal(0,5),

with λX[i]indicating the average coefficient across all odorants, and σλX > 0.

Bayesian within-subject analysis for color response reproducibility

The participants repeated the associated color response twice on different days. We performed Bayesian analysis to estimate the mean difference in color responses between days to demonstrate the reproducibility of the color responses among the participants.

We modeled the difference in a*-axis values measured from the associated color responses as a normal distribution as follows:

Δa[i,j]∗∼Normal(μodor_a[i],σodor_a[i]),

μodor_a[i]∼Normal(μall_a,σall_a),

σodor_a[i]>0,

σall_a>0,

where i indicates the odor ID, j denotes the data index, and Δa[i,j]∗ indicates the difference between the a*-value on day 2 and day 1 for each participant and each odorant. μodor_a[i] indicates the mean difference of Δa[i,j]∗ within an odor and σodor_a[i] indicates its standard deviation. μodor_a[i] followed a normal distribution with a mean difference across the odors, μall_a, and a standard deviation, σall_a. Moreover, we modeled the difference in the b*- and L*-axis values between days as Δa[i,j]∗.

Results

Days-differences of associated colors within participants

Perception from olfaction could vary among participants. We analyzed whether the responses derived from the odorants were consisted among participants. The conventional common approach cannot support the hypothesis that “the mean of differences was equally 0.” To confirm within-participant consistency with the responses derived from the odorants, we estimated the posterior distribution and its 95% Bayesian confidence interval (CI) of the mean day differences within the participants. For the a*-, b*-, and L*-axis values, there were no parameters of day difference whose 95% CI did not include 0 (Fig. 2). The associated color responses did not significantly fluctuate with the days in response to the odorant. The results indicated that the associated color responses did not significantly fluctuate with the days in response to any odorant.

Figure 2 Color reproducibility between the different days in (A) a*-axis, (B) b*-axis, and (C) L*-axis values.

Gray bars indicated the 95% Bayesian confidence interval. Black circles indicate estimated Bayesian mean values.

Confirmation of the multicollinearity for multilevel regression

Before implementing the multilevel regression, we confirmed the multicollinearity of five descriptive ratings. Subsequently, we performed Pearson correlation analysis to demonstrate the correlation coefficients and any absolute values of the coefficients ≤0.70 (Table 2), which is the lower limit of a strong correlation in psychology (Akoglu, 2018). We included all descriptive rating data in the multilevel regression model according to the results.

Table 2 Pearson’s correlation coefficients among the descriptive ratings.

	Strength	Pleasantness	Familiarity	Edibility	Arousal	
Strength	–	−0.053	0.20	0.48	−0.0030	
Pleasantness		–	0.45	0.12	0.45	
Familiarity			–	0.21	0.50	
Edibility				–	−0.087	
Arousal					–	

Bayesian multilevel regression

First, we confirmed the model for a*-value estimation and the distribution of the coefficient parameters with odor descriptors. A significantly positive estimated parameter with ratings of a semantic descriptor indicated that a higher descriptor rating would allow the associated colors to be reddish. By contrast, a significantly negative estimated parameter suggested that a higher descriptor would allow the associated colors to be greenish.

The multilevel model for a*-values estimated the parameters with odorant-level effects. We used a multilevel Bayesian regression model that included random intercepts and slopes for each odor to estimate the effect of each descriptive rating on the associated colors. We estimated the coefficients of the five descriptive ratings, and their 95% CIs are demonstrated for each odorant (Fig. 3). For the 95% CIs of Edibility, the three odors, including 2-ethyl pyrazine, limonene, and strawberry aldehyde, did not significantly overlap with zero (Fig. 3). The results indicated colors associated with the three odors related to Edibility ratings. An increase in the Edibility ratings of these odorants would make the associated colors more reddish. The remaining odors did not display a practical difference in the coefficients of Edibility from zero in the a*-value predictive model. The remaining descriptive ratings did not demonstrate any practical differences from zero for any odor.

Figure 3 The mean and 95% confidence interval (CI) of the coefficients of Edibility, Strength, Familiarity, Pleasantness, and Arousal estimated by the Bayesian multilevel regression model for the a*-value prediction.

The black vertical line indicates 0. The black dots indicate the Bayesian mean, and bold bars indicate 95% Bayesian CI. The red bars indicate 95% significant coefficients, and the gray bars indicate non-significant coefficients.

Figure 4 depicts the colors associated with the three odors that showed higher coefficients with Edibility ratings. The color responses were reddish, according to the increase in Edibility ratings for the three odorants (Fig. 4). The original colors indicated the native responses of the participants. To demonstrate the relationship between the a*-axis values and Edibility ratings, we fixed the b*- and L*-parameters with the initial settings. Subsequently, we changed the a*-values only according to the responses of the associated colors (Fig. 4, middle row). The right row denotes changes in the a*-value (Fig. 4). The associated colors were greenish and reddish for low and high Edibility ratings in the three odors, respectively. These a*-value changes corresponded to the Edibility ratings.

Figure 4 Plots of associated colors selected by each participant for significant odors in the regression model for a* prediction.

(left) The selected colors are displayed along with the edibility ratings. (middle) The colors are modified to denote changes on the a*-axis in the CIE L*a*b* space. The a* values reflect the individual responses of the associated colors. The colors of b* and L* values are fixed (L* = 70, b* = 38; refer to Materials and Methods). The colors are reddish for higher a*-values. (right) The a* values of the selected colors. The color bars indicate the range of a*-values.

Consequently, we confirmed the model that estimated the b* values and the distribution of estimates of the coefficient parameters with their odor descriptors. A significantly positive estimated parameter with ratings of a semantic descriptor indicated that a higher descriptor rating would allow the associated colors to be yellowish, whereas a lower rating would allow the colors to be bluish. An estimated parameter that was significantly negative from 0 suggested that a higher descriptor would allow the associated colors to be bluish along the yellow-blue axis.

For the Edibility ratings, five odorants, including hydroxy citronellal, butyl acetate, (1S)-(-)-alpha-pinene, trimethyl amine (30%), and 2,3-Dimethylpyrazine, displayed significant coefficients. The 95% CI of the five odors did not significantly overlap with zero (Fig. 5). In the results of the a*-prediction model, the five odors did not reveal significant coefficients with Edibility (Fig. 3). Figure 6 represents the color responses from the participants with Edibility ratings. To demonstrate the relationship between the b*-axis values and Edibility ratings, we fixed the a*- and L*-parameters with the initial settings. Moreover, we changed the b* values only according to the responses of the associated colors (Fig. 6, middle row). The associated colors were bluish for low Edibility ratings.

Figure 5 The mean and 95% confidence interval (CI) of the coefficients of Edibility, Strength, Familiarity, Pleasantness, and Arousal estimated by the Bayesian multilevel regression model for b*-value prediction.

The black vertical line indicates 0. The black dots indicate the Bayesian mean, and bold bars indicate 95% Bayesian CI. The red bars indicate 95% significant coefficients, and the gray bars indicate non-significant coefficients.

Figure 6 Plots of the associated colors selected by each participant for significant odors in the regression model for b* prediction.

(left) The selected colors are displayed along with the Edibility ratings. (middle) The colors are modified to denote changes on the b*-axis in the CIE L*a*b* space. The b* values reflect the individual responses of associated colors. The colors of a* and L* values are fixed (L* = 70, a* = 20; refer to Materials and Methods). The colors are yellowish for higher b*-values. (right) The b* values of the selected colors. The color bars indicate the range of b*-values.

For Arousal ratings, the two odorants, namely hydroxy citronellal and 4-methyl-3-penten-2-one, displayed higher coefficients (Fig. 7). The associated colors would be more bluish for these odorants following low Arousal ratings. The remaining descriptive ratings did not reveal any practical differences from zero for any odor.

Figure 7 Plots of the associated colors selected by each participant for significant odors in the regression model for b* prediction.

(left) The selected colors are displayed along with the Arousal ratings. (middle) The colors are modified to denote changes on the b*-axis in the CIE L*a*b* space. The b* values reflect the individual responses of associated colors. The colors of a* and L* values are fixed (L* = 70, a* = 20; see Materials and Methods). The colors are yellowish for higher b*-values. (right) The b* values of the selected colors. The color bars indicate the range of b*-values.

The model for L*-value prediction demonstrated significant coefficients for Strength and intercepts for all odorants. The coefficients of Strength ratings indicated that the associated colors would be darker for all odorants following an increase in ratings (Fig. 8). We displayed some examples to exhibit the relationship between the Strength ratings and the lightness of colors (Fig. 9), which revealed that the colors would be darker for higher ratings.

Figure 8 The mean and 95% confidence interval (CI) of the coefficients of Strength, Edibility, Familiarity, Pleasantness, and Arousal estimated by the Bayesian multilevel regression model for L*-value prediction.

The black vertical line indicates 0. The black dots indicate the Bayesian mean, and bold bars indicate 95% Bayesian CI. The red bars indicate 95% significant coefficients, and the gray bars indicate non-significant coefficients.

Figure 9 Example plots of the associated colors selected for each odor by each participant.

(left) The selected colors are displayed along with the Strength ratings. (right) The L* values of the selected colors. The color bars indicate the range of L*-values. The figure denotes two examples of odors and the lightness of the associated colors; however, all odors indicate significant effects on the Strength of the L* values (refer to Fig. 8).

Discussion

There were two purposes of this study. The first aim was to identify the descriptive evaluations of odors to predict the associated colors, while the second aim was to predict the influence of the ratings on hue and lightness of the associated colors, accounting for the differences in the odor types. The results demonstrated that the Edibility and Arousal ratings were related to the color hues with several odors, represented by the a*- and b*-axes. Based on the odor type, the effects of Edibility on color hue are represented in different hue axes, namely a* or b*. The Strength of the odor was related to the color lightness in any odorant. The descriptive ratings of Edibility and Arousal mediated the hues of associated colors for specific odors, and odor Strength generally mediated the lightness of the associated colors.

First, we will discuss the effects of Edibility on odor-color associations. The multilevel regression model using Bayesian estimation revealed that Edibility and Arousal ratings were related to some odor-color associations in the color hue. The positive values in the Edibility coefficients are presented in three odors for the a*-regression model. The associated colors of these three odors were reddish for higher Edibility ratings. The Edibility coefficients for b*-regression demonstrated significant positive coefficients for the five odorants. The positive coefficients in the b*-model indicated that the associated colors of the five odorants were yellowish following an increase in Edibility. The specificity of the Edibility coefficients with the two color-hue regression models, i.e., a*- and b*-models in CIE L*a*b* space, resulted in the following two suggestions: (1) the descriptor of Edibility can mediate the associated colors with various odorants (in this case, eight of 13 odorants) and (2) the color-associations were mediated by the Edibility ratings in either of two axes, namely a*- or b*-in the CIE L*a*b* color space. The associated colors are not simultaneously mediated by the a*- and b*-values. These speculations were consistent with a report that demonstrated that the odors with higher edibility tended to be yellow colors (Maric & Jacquot, 2013). On the other hand, for strawberry odors, several studies reported conflicting associated colors, red or pink (Dematte, Sanabria & Spence, 2006; Osterbauer et al., 2005); however, green was reported in an Australian study (Stevenson, Rich & Russell, 2012). In our analysis, strawberry aldehyde was associated with reddish colors when the odor was evaluated highly edible. Moreover, the odor from almond essence was reportedly associated with blue color (Stevenson, Rich & Russell, 2012); however, our model in 2-ethyl pyrazine, which is the main component of burned nuts, demonstrated reddish colors when highly edible. The varied association could be influenced by the difference in the regions and cultural backgrounds (Jacquot et al., 2016; Levitan et al., 2014; Nehmé et al., 2016) between Western and Japanese people in East Asia. Cultural backgrounds can vary in the olfactory perception, recognition performance (Chrea et al., 2004), and pleasantness of odors (Herz, 2005).

Our results on the effects of Edibility on odor-color associations were supported by conventional studies, which have proposed the importance of edibility in odor description and olfactory discrimination in terms of psychological and neurophysiological aspects. For example, in the odor naming task of Dutch words, odors that were evaluated as “edible” were correctly labeled more frequently than those that were considered of “low edibility” (Huisman & Majid, 2018). In the neuronal activity in the olfactory system, edibility (or toxicity as the flip side of edibility) is one of the primary dimensions of human odor perception (Haddad et al., 2010). Smeets & Dijksterhuis (2014) have reported on the efficacy of food odors as goal primes, such as digestion and appetite regulation. Gustatory sensation can be affected other sensory inputs. Oleszkiewicz et al. (2023) reported that individuals with blindness and deafness showed different taste sensitivity and liking, probably owing to less gustatory experiences. Functional MRI studies have demonstrated that the presentation of congruent odor-color pairs induces higher activity in the orbitofrontal cortex, whose activity diminishes by satiety, than incongruent odor-color combinations (Osterbauer et al., 2005). Our finding that the odor-color association was mediated by the Edibility rating was consistent with these studies.

Regarding the b*-values in the blue and yellow axes in the CIE L*a*b* space, Arousal ratings displayed higher coefficients for two odorants, namely hydroxy citronellal and 4-Methyl-3-penten-2-one. These odors may have simultaneously activated trigeminal sensations with olfactory sensations owing to the olfactory features of these odorants. According to the TGSC Information System, the odor types of hydroxy citronellal were “fresh,” and those of 4-Methyl-3-penten-2-one were “pungent.” One probable interpretation is that these odorants activate the intranasal chemosensory trigeminal system. The trigeminal sensation induces arousal feelings, and the Arousal rating may affect the b*-values. These results suggested that trigeminal sensation induced a yellowish color. Michael & Rolhion (2008) reported that some colors can relate to specific nasal thermal sensations. However, we could not determine a relationship between the color association and nasal sensation in the present study. This necessitates further investigation to determine the association between yellowish color and arousal.

Although several odors showed significant relationships with a*- or b*-values in the models as discussed above, the remaining odors did not demonstrate any significant relationship between the associated color hues and present descriptors. Our findings could not conclude that the descriptive ratings of odors did not mediate the pairs of odor-color associations. The present experiment investigated only five descriptors of odor impressions. Researchers should assess other candidates for descriptive ratings to determine if these ratings relate to the parameters of the associated colors.

In the L*-axis of lightness, the Strength rating was negative from 0 for all the tested odorants. Thus, the associated colors of the tested odorant were darker upon evaluating the odorants as strong. This finding was consistent with those of a previous study that reported on an inverse relationship between odor intensity and color lightness (Kemp & Gilbert, 1997). We confirmed that odor intensity can globally modulate the lightness of the associated color using our procedure and analyses.

This study introduced a Bayesian multilevel regression model to predict the effects of descriptive ratings on the parameters of the associated colors. This multilevel model enabled the incorporation of odor-specific effects that varied across odor types. Furthermore, such models can represent complex structures by assuming conditional dependencies. The Bayesian multilevel model did not begin with the assumption that the null hypothesis was true; therefore, this method did not consider type-1 error from multiple comparisons, as do conventional statistical methods (Gelman, Hill & Yajima, 2012). The present multilevel model included random slopes and intercepts among the odors, without multiple comparisons. Using the aforementioned model, we estimated each odor parameter derived from the common parameters of all odors. This estimation could represent the difference in odors with descriptive ratings using the identical model.

Our measurement method had several limitations. Our experiment determined the response colors within a circle in the CIE L*a*b* with a fixed radius. It aimed to prevent manipulative confusion and errors during the response to the associated colors. For further investigation, researchers should measure the associated colors using free radius of color hues and a continuous degree of lightness. Furthermore, the measurement of descriptive ratings requires improvement. The study warranted higher linguistic descriptors to elucidate the relationship between descriptive expression and odor-color association; nonetheless, our findings confirmed that the Edibility rating of odorants could mediate odor-color associations. Future studies should focus on variable food-related descriptors and odor-color associations with continuous measurements.

We do not enforce that the tested odor should be associated with the colors as demonstrated in our manuscript (e.g., 2-ethyl pyrazine and reddish colors). Our models aimed to investigate the influence of olfactory descriptive ratings on the feature of associated colors. Furthermore, investigation was only performed in individuals with a similar cultural background in Japan. Cross cultural difference comparison would be possible if the proposed method is used for other populations with different cultural backgrounds.

Conclusions

In summary, we investigated the impact of descriptive ratings on the associated colors from odors using the Bayesian multilevel regression model. Our analysis could predict the olfactory description that would be useful to expect an associated color for each odor. The results indicated that their Edibility or Arousal ratings mediated the hue values of the associated colors of some odors. In contrast, the Strength of odors generally mediated the lightness of their associated colors. Our investigation and measurement procedures will contribute to the understanding of odor-color association.

Supplemental Information

Supplemental Information 1 Bayesian multilevel regression models to estimate the sex differences of color responses using odor-level effects.

Click here for additional data file.

Supplemental Information 2 Multilevel regression analysis including intercepts from individual errors.

Click here for additional data file.

Supplemental Information 3 Sex differences of the odor-color responses in (A) a*-axis, (B) b*-axis, and (C) L*-axis values.

Gray bars indicate the 95% Bayesian confidence interval. Black circles indicate the estimated Bayesian mean values.

Click here for additional data file.

Supplemental Information 4 The mean and 95% confidence interval (CI) of the coefficients estimated by the Bayesian multilevel regression model including individual errors for a*-value prediction.

The black vertical line indicates 0. The black dots indicate the Bayesian mean, and bold bars indicate 95% Bayesian CI. The red bars indicate 95% significant coefficients, and the gray bars indicate non-significant coefficients.

Click here for additional data file.

Supplemental Information 5 The mean and 95% confidence interval (CI) of the coefficients estimated by the Bayesian multilevel regression model including individual errors for b*-value prediction.

The black vertical line indicates 0. The black dots indicate the Bayesian mean, and bold bars indicate 95% Bayesian CI. The red bars indicate 95% significant coefficients, and the gray bars indicate non-significant coefficients.

Click here for additional data file.

Supplemental Information 6 The mean and 95% confidence interval (CI) of the coefficients estimated by the Bayesian multilevel regression model including individual errors for L*-value prediction.

The black vertical line indicates 0. The black dots indicate the Bayesian mean, and bold bars indicate 95% Bayesian CI. The red bars indicate 95% significant coefficients, and the gray bars indicate non-significant coefficients.

Click here for additional data file.

Additional Information and Declarations

Competing Interests

Author Contributions

Human Ethics

Data Availability

The authors declare that they have no competing interests.

Kaori Tamura conceived and designed the experiments, performed the experiments, analyzed the data, prepared figures and/or tables, authored or reviewed drafts of the article, and approved the final draft.

Tsuyoshi Okamoto conceived and designed the experiments, authored or reviewed drafts of the article, and approved the final draft.

The following information was supplied relating to ethical approvals (i.e., approving body and any reference numbers):

The local ethics committee of the Faculty of Arts and Science, Kyushu University approved the current experiment. All procedures were performed in accordance with the approved guidelines of the local ethics committee of the Faculty of Arts and Science, Kyushu University.

The following information was supplied regarding data availability:

The data and code are available at Zenodo: Kaori Tamura. (2023). Data and codes of “Odor descriptive ratings can predict some odor-color associations in different color features of hue or lightness” (https://gitlab.com/tamurak415/olfqr) [Data set]. Zenodo. https://doi.org/10.5281/zenodo.7693843.

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
