# Peer review of "Odor descriptive ratings can predict some odor-color associations in different color features of hue or lightness"

_PeerJ, doi:10.7717/peerj.15251_

## Round 0.1 · original submission · Major Revisions

While there were some positive comments from the reviewer, there are some major issues that the authors must address. A significant concern on the part of the reviewers is in the references, where some apparently significant references are not included and other references are not necessarily related to the issues under discussion when those references appear in the paper.

Perhaps the most concerning issue is the small sample size. The authors should address how the sample size was established (presumably accessibility?) and should compare their sample size to a post-hoc power analysis.

In addition to the sample size, the other major area of concern is regarding the following comment from reviewer #3: "A fundamental problem with this paper is that it claims that odor descriptions affect odor-color associations, but odor descriptions were not elicited or manipulated. Instead, ratings of odor dimensions were collected". It is therefore unclear exactly what the authors intended to do or what gap in knowledge the authors intended to fill, and whether the study addressed the problem that the authors set out to examine. While the authors are invited to submit a revision, changes will need to be significant in terms of explaining the motivation for the study and how the study fits with the already-established literature. Please address every comment from the reviewers in detail.

Reviewer 1 ·

Basic reporting

The Manuscript entitled “Odor descriptions can mediate some odor-color associations in different color features of hue or lightness” evaluate an interesting topic. The study aimed to evaluate the odor descriptive attributes that mediate odor-color information.
In general, the Manuscript is well written, newsworthy and clear to understand, but requires some major revisions. In particular, the Manuscript showed some flaws and requires a careful revision by Authors in Introduction, Methods, Results, and Discussion.

Experimental design

My major concern is the low number of subjects enrolled in the study.Authors should perform other studies in order to increase the number of participants (11 women and 12 men). Usually in the literature were reported sex differences in odor perception. This finding may represent a bias in the statistical evaluations.
Methods: Please add some information if participants stopped drinking and eating and if they wash their mouth before the odour discrimination. In fact, some odours can be perceived from retro nasal cavity.

Validity of the findings

The Introduction section is too long and it should be focused on the specific topic. Authors should revise the Introduction section in order to obtain a better comprehension of the text.
Line 40 Please substitute Dematte et al. With Dematte & Colleagues
Line 42: Change smells with smell
Line 60: please delete italic and capital letter for Pleasantness
Line 278 Please delete double brackets.
Line 345: Please delete “ from "Edibility".
Line 354: Please correct Michael et al. with Michael & Rolhion (2008) and delete the reference at the end of the sentence.

Line 365 Authors write “the Strength rating was negative for all odorants. Thus, the associated colors of any odorant were darker upon evaluating the odorants as strong.” Did Authors try different concentration of odors?
In the Discussion section authors should include the following reference as regards associations color-odors:
Oleszkiewicz et al., 2022. https://doi.org/10.1016/j.foodqual.2022.104712
Table 2: please delete the last line
Figure 1: please correct 20-mL, insert the point after odorant and substitute Subsequently with Second.

Reviewer 2 ·

Basic reporting

Generally speaking, reporting was appropriate. However, I do have some recommendations:

The authors mention that the study was done twice on two different days. Maybe I missed this, but is there a section where they report on stability over time? I'd like to see how people's responses are correlated/whether they changed much (and also to know whether the order was different within the two sessions; line 130 is a bit unclear on this point with the use of the word "among").

Smaller notes:
line 20: Do you perhaps mean individual differences in odor associations? Just saying individual differences in odors is too broad.
line 30: Where it says "remaining" is confusing as there were 13 odors, and 3 were associated with reddish, so there are more than 5 remaining.
line 31-33: The final sentence restates what has just been said. It can be omitted (and perhaps if this frees up space, a sentence of implications could be a good addition to the abstract).
line 145: "inquired" seems to the be wrong word here
line 149: a circle within the color space -- the center is specified but not the angle. Could a figure show an approximation of the circle itself?
line 155: Were there any anchors on the scales or given verbally in instruction to the participants? In general a little more info on the ratings would be helpful to get the full connotation of what the descriptors meant in Japanese. I do wonder how people understood the scales -- for example, if one finds an odor to be not at all pleasant, but not unpleasant, does that get marked in the center of the scale or at the far left?
Figure 1a -- typo for "coffee" in the image itself
Figure 2 and later figure captions: a reminder of what negative means in this context could be helpful as that is rather buried in the paper

Experimental design

Overall the design seemed appropriate. The research question is certainly suitable for this journal.

The rationale for this study's methods seems to be, in part, that using the color wheel avoids the priming. However, with the wheel, do the participants see no color until they click somewhere on the wheel? Did they see the full circle (doesn't seem like it in the figure)? Couldn't participants be primed by where they happen to click? This doesn't seem like a particularly compelling knowledge gap, but if this is in fact the motivating question then the discussion could more directly come back to this and explicitly compare findings to those taken using a color patch approach.

The abstract rationale for the knowledge gap has to do with odor types, but I also didn't see this theme well carried through. I am unsure of what the "seven types" of odors were. Perhaps Table 1 could show which orders are in which "type" and the text could more clearly address this/show which analyses look at type and what they found.

Validity of the findings

In general, the discussion/conclusion could more directly address how these results compare to other studies. For example, when discussing edibility (lines 344-345), the authors note that their work is consistent with other studies that find a role for edibility. But does the interaction follow the same pattern of association? For the L*a*b* there was a bit more of this, but I'd like it to all be quite explicit and easy to follow so that I can understand whether these results show the same relationships as past color-odor work. If they do/don't, does that allow the researchers to say more about the different hypotheses underlying these relationships (e.g., intensity/edibility could work similarly across different samples or if they are mediated by culture/language then this Japanese sample might show different relationships than the samples in the studies discussed in the introduction).

Reviewer 3 ·

Basic reporting

• The article is missing citation of the work on Charles Spence. He has published many studies on this topic so should be cited.
• Often the same concept is referred to in multiple different ways which makes it difficult to follow e.g., descriptive impressions, descriptive attributes, descriptive ratings. Please stick to one term throughout.
• In general, the introduction does not motivate the study well. Many aspects are stated, but it is difficult to connect the points made across paragraphs, and to understand what the research gap is.
• The explanation for why odor-color associations exist is unclear in the text.
• In general there are many examples of ambiguous phrasing throughout the introduction e.g., “Researchers have discussed ways to build a specific linkage to color and odor” – it’s unclear what this means.
• References are used incorrectly (i.e., papers cited that didn’t research the topic described)
• The odor dimensions investigated are not motivated.

Experimental design

• A fundamental problem with this paper is that it claims that odor descriptions affect odor-color associations, but odor descriptions were not elicited or manipulated. Instead, ratings of odor dimensions were collected.
• The sample size is not justified. 23 participants seems small.
• A strength of the design is that participants did the task twice across 2 days, which allows one to test for consistency of associations.
• It is not clear why participants could only choose one of three levels of lightness and not instead use a continuous value?
• How can a VAS scale run from 0 to 1? How many points were on the scale?
• As far as I can tell, the analysis only looks at odor-level data and not participant data. The analysis should take both into account.
• The authors say they “confirmed” multicollinearity but did not explain how they dealt with this.
• It is not clear to me why an odor would be considered more red versus green and why this would be theoretically interesting.

Validity of the findings

• Because the study is not well-motivated I cannot understand what theoretical impact it would have.

Additional comments

• While I am sympathetic to the broad goal of understanding what dimensions underlie crossmodal odor associations, the specific associations here are not well-motivated or well-explained.
• Throughout the paper the authors refer to “descriptions”, but descriptions are not investigated in this study. Ratings for odor dimensions are collected.
• The abstract should end with an overall conclusion.
• It would be helpful to describe the de Valk et al. study in more detail.
• Majid and Burenhult (2014) and Majid et al. (2018) did not look at odor-color associations.

---

## Round 0.2 · accepted · Accept

The authors have addressed the issues raised by the reviewers. Please address this comment from the reviewer in the final version of the manuscript:

> Line 59 Authors should delete comma at the end of the sentence.

Reviewer 1 ·

Basic reporting

The Manuscript should be accepted in the present form

Experimental design

The Experimental design is well-documented and easy to understand.

Validity of the findings

Line 59 Authors should delete comma at the end of the sentence.